# Substituent Effects of the Nitrogen Heterocycle on Indole and Quinoline HDN Performance: A Combination of Experiments and Theoretical Study

**DOI:** 10.3390/ijms24033044

**Published:** 2023-02-03

**Authors:** Shujiao Jiang, Sijia Ding, Yasong Zhou, Shenghua Yuan, Xinguo Geng, Zhengkai Cao

**Affiliations:** 1SINOPEC Dalian Research Institute of Petroleum and Petrochemicals Co., Ltd., Dalian 116041, China; 2State Key Laboratory of Heavy Oil Processing, China University of Petroleum, Beijing 102249, China

**Keywords:** HDN, indole, quinoline, substituent effects, DFT

## Abstract

Hydrodenitrogenation (HDN) experiments and density functional theory (DFT) calculations were combined herein to study the substituent effects of the nitrogen heterocycle on the HDN behaviors of indole and quinoline. Indole (IND), 2-methyl-indole (2-M-IND), 3-methyl-indole (3-M-IND), quinoline (QL), 2-methyl-quinoline (2-M-QL) and 3-methyl-quinoline (3-M-QL) were used as the HDN reactant on the NiMo/γ-Al_2_O_3_ catalyst. Some key elementary reactions in the HDN process of these nitrogen compounds on the Ni-Mo-S active nanocluster were calculated. The notable difference between IND and QL in the HDN is that dihydro-indole (DHI) can directly convert to O-ethyl aniline via the C–N bond cleavage, whereas tetrahydro-quinoline (THQ) can only break the C–N single bond via the full hydrogenation saturation of the aromatic ring. The reason for this is that the –NH and C=C groups of DHI can be coplanar and well adsorbed on the Ni-Mo-edge simultaneously during the C–N bond cleavage. In comparison, those of THQ cannot stably simultaneously adsorb on the Ni-Mo-edge because of the non-coplanarity. Whenever the methyl group locates on the α-C or the β-C atom of indole, the hydrogenation ability of the nitrogen heterocycle will be evidently weakened because the methyl group increases the space requirement of the sp^3^ carbon, and the impaction of the C=C groups on the Ni-S-edge cannot provide enough space. When the methyl groups are located on the α-C of quinoline, the self-HDN behavior of 2-M-QL is similar to quinoline, whereas the competitive HDN ability of 2-M-QL in the homologs is evidently weakened because the methyl group on the α-C hinders the contact between the N atom of 2-M-QL and the exposed metal atom of the coordinatively unsaturated active sites (CUS). When the methyl group locates on the β-C of quinoline, the C–N bond cleavage of 3-methyl-quinoline becomes more difficult because the methyl group on the β-C increases the steric hindrance of the C=C group. However, the competitive HDN ability of 3-M-QL is not evidently influenced because the methyl group on the β-C does not evidently hinder the adsorption of 3-M-QL on the active sites.

## 1. Introduction

Hydrotreating is a current industrialized process used in clean fuel production [1,2,3]. Hydrodenitrogenation (HDN) is an important reaction in the hydrotreating process [4,5]. The nitrogen compounds will severely impair the quality of petroleum production or the processability of the oil fractions; therefore, the content of nitrogen usually needs rigid control [6,7,8,9,10]. Similar to hydrodesulfurization (HDS), the substituent groups on the nitrogen compounds evidently influence the HDN behavior in the model compounds and the actual oil fractions [11,12,13]. The substituent effects on the HDS have been amply studied. It is well acknowledged that the substituent groups close to the S atom of the thiophene compounds (such as 4,6-dimethyl-dibenzothiophene) could result in a more severely negative influence on the HDS activity [14]. The substituent groups near the S atom could influence the adsorption and the C–S bond cleavage on the Ni-Mo-S active nanocluster [15,16,17]. Theoretically, the electron distributions and the hydrogenolysis mechanisms of the sulfur compounds, the basic nitrogen compounds and the non-basic nitrogen compounds are all different [18,19,20,21,22,23,24,25,26]. Therefore, the substituent effects of the basic and the non-basic nitrogen compounds are worthy of a contrastive study. Currently, quinoline is used as the typical basic nitrogen compound, and indole is the non-basic nitrogen compound in HDN research [27,28,29]. Some studies have proposed that the substituent groups on the heterocycle will notably influence the HDN behavior of the nitrogen compounds rather than those on the aromatic rings [30,31]. Therefore, the substituent effects were focused on the heterocycle of quinoline and indole in this study.

In addition to the experiments, quantum chemical calculations with density functional theory were another technique used to understand the HDN process. Previous studies predicted that the atomic structure of the original and Co/Ni promoted MoS_2_ nanoclusters [32,33,34,35], adsorption of the nitrogen compounds [36] and conversion of the complicated sulfur compounds catalyzed on the various active sites [37,38,39,40]. Based on these theoretical studies, calculations of the HDN process on the Ni-Mo-S active nanocluster are technically feasible. In this study, DFT calculations were used to predict and explain the experimental differences resulting from the ring structure and the substituent position in the indole and quinoline HDN process. A better understanding of the substituent effects of the indole and quinoline on the HDN behaviors will be helpful for the design of hydrotreating catalysts.

## 2. Results and Discussion 

### 2.1. HDN Behaviors of Indole and Quinoline on the Ni-Mo-S/γ-Al_2_O_3_ Catalyst

#### 2.1.1. Experimental Phenomenon

The HDN behaviors of indole and quinoline on sulfurized Ni-Mo-S/γ-Al_2_O_3_ were investigated at first. The HDN products of the indole and quinoline are listed in Table 1. The relative contents of the indole and quinoline HDN products at various liquid hourly space velocities (LHSVs) are shown in Table 2. There are two points that need to be noted: First, the fast hydrogenation saturation of the nitrogen heterocycle for both indole and quinoline. The contents of DHI and THQ-1 already reached a peak at 20.0 h^−1^ LHSV. This fast conversion was explored via a theoretical study, and it was seen that the substituent groups influenced this process in some cases. Second, the contents of OEA and OPA were quite different. The content of OEA could exceed 18% during the indole HDN, whereas the content of OPA was always no more than 1% during the quinoline HDN, indicating that the conversion from THQ-1 to OPA was slow [41]. This phenomenon can be used to predict the principle of the C–N bond cleavage on the Ni-Mo-S active sites.

#### 2.1.2. Theoretical Explanation

DFT calculations were used to explain the reasons for the above two points. The first step in the indole and quinoline HDN is the hydrogenation saturation of the nitrogen heterocycle. The reaction pathway from IND to DHI is short because there are only two active hydrogen atoms participating. This hydrogenation process catalyzed on the Ni-S-edge and Ni-Mo-edge of the Ni-Mo-S active nanocluster is listed in Table 3. The ΔE in this table stands for the energy difference between the current state and the previous state. For instance, the ΔE value corresponding to TS-3 is the energy change from the monohydrogen-indole to this transition state, namely, the activation energy of the reaction from monohydrogen-indole to DHI. The adsorption of indole on the active sites relies on the C=C group or the conjugate π electrons of the nitrogen heterocycle [37]. The hydrogen transfer from the active sites to the unsaturated carbon atoms is easy, and the activation energy is smaller than 130 kJ·mol^−1^. The main difference between the Ni-S-edge and Ni-Mo-edge is that the –SH group can provide steady existing active hydrogen on the Ni-S-edge, whereas the Ni-Mo-edge has to dissociate the hydrogen molecule in the reaction atmosphere with higher activation energy before the hydrogen transfer. Moreover, this hydrogen dissociation is unstable, and the active hydrogen atoms are likely to merge and return to the reaction atmosphere. It can be considered that the conversion from indole to DHI occurs more easily on the Ni-S-edge, and the substituent effects of indole can influence this conversion (see Section 3.2). 

The conversion from QL to THQ consists of more elementary reactions than that of IND (as shown in Table 4). The adsorption of quinoline relies on the lone pair of electrons of the nitrogen atom [37], and this adsorption is stronger than that of indole on the same active center. The hydrogen transfer from the –SH of the Ni-S-edge to the quinoline N atom needs to overcome a +140.22 kJ·mol^−1^ energy barrier because the interaction between the N atom and the exposed metal is strong. On the other hand, as the quinoline adsorption morphology on the Ni-Mo-edge is vertical, the hydrogen molecule can be dissociated on the Ni–Mo atom pairs and the activation energy of this dissociation pathway is much lower than that on the Mo–S atom pairs. Despite the hydrogenation saturation of quinoline both on the Ni-S-edge and the Ni-Mo-edge having a rate control step, the reaction barrier is, at most, 140 kJ·mol^−1^. Therefore, it can also be considered that this conversion could be carried out smoothly on the Ni-Mo-S active nanocluster without the obvious regional selectivity. 

As mentioned above, the content of OEA in the indole HDN production is considerable, whereas the amount of OPA for the quinoline HDN is small. This phenomenon can be explained by the C–N bond cleavage on the Ni-Mo-S active sites. There are two possible reaction pathways of the C–N bond cleavage, namely, the S_N_2 and the E_2_ pathways because the Ni/Co-promoted Mo-edge is the primary active site for the hydrogenolysis [15,42]. The C–N bond cleavage on the active sites of the Ni-Mo-S edge is discussed in detail in this section. The comparison of the C–N bond cleavage via S_N_2 and E_2_ is based on the process for DHI to OEA. The conversion on the Ni-Mo-edge is listed in Table 5. According to the result of the calculation, the hydrogenolysis of the C–N bond via the S_N_2 route is an endothermic process, and the activation energy exceeds 260 kJ·mol^−1^. The activation energy is much higher than the C–S bond cleavage on the same active site [15,43]. 

For an elementary reaction, the activation energy is the energy difference between the reactant and the transition state. The activation energy is mainly determined by two factors: the bonding energy change and the adsorption energy change from the reactant to the transition state. In the transition state, the distance between the N and the α-C atoms exceeds 3.0 Å, indicating that the C–N bond of DHI has completely cleaved. The molecular structure and the adsorption difference between DHI and its C–N bond cleavage product are listed in Table 6. The cleavage of the C–N bond results in an approximately 280 kJ·mol^−1^ bonding energy elevation, and this is the main contribution to the activation energy. Despite the intermediate of the C–N bond cleavage having two molecular orbitals with low eigenvalues, the adsorption morphologies of DHI and its hydrogenolysis intermediate are quite similar. The newly formed molecular orbital of the intermediate (eigenvalue: −5.76 eV) is blocked by the substituent group on the benzene ring. The N atom on the hydrogenolysis intermediate cannot bond with the N and Mo atoms simultaneously. Therefore, the adsorption energy difference between DHI and its hydrogenolysis intermediate is only approximately 100 kJ·mol^−1^. This difference cannot compensate for the bonding energy elevation caused by the C–N bond cleavage, and this is the reason why the S_N_2 route is not likely to be the dominant reaction pathway for the C–N bond cleavage.

The E_2_ route is also considered a possible pathway for the C–N bond cleavage. The hydrogen atom on the β-C needs to transfer to the active sites first, and the DHI converts to monohydroindole (MHI). Then, the C–N bond cleaves, forming a -NH group and C=C double bonds. As the concentration of DHI in the product is quite high, it can be predicted that the DHI can desorb and relocate on the active sites. When the β-C of DHI comes close to the Mo–S atom pair, the hydrogen on the β-C can transfer to the sulfur atom on the active site, and the β-C comes close to the Mo atom at the same time. The activation energy of this conversion is only 120–140 kJ·mol^−1^, and the reaction energy is also smaller because the β-C atom of MHI is attached to the Mo atoms, which could decrease the system energy. 

The second step of the elimination pathway is the break of the C–N bond. As the C–N bond cleaves, the β-C and the aromatic carbon attach with the Mo atom on the active sites. Meanwhile, the β-C atom forms a conjugated system with the aromatic ring, which also further lowers the system energy. In addition, the deformation of the active sites in the second step is slight. Especially, there is no active hydrogen transfer indicating that the bonding energy’s elevation at the transition state only comes from the self C–N bond cleavage. Therefore, the activation energy for the second step is only 130–140 kJ·mol^−1^, which is much lower than the C–N cleavage via the hydrogenolysis pathway. It can be indicated that the elimination pathway divides the hydrogenolysis pathway into two steps with much lower activation energies, and it is probably the dominant pathway for the C–N bond cleavage on the nitrogen heterocycle.

The DHI can directly convert to OEA, whereas the conversion from THQ-1 to OPA is not likely to be the dominant pathway of the C–N bond cleavage in the HDN experiments. Similar to DHI, THQ-1 can also cleave the C–N bond via the elimination reaction (as shown in Table 7). For THQ-1, the hydrogen transfer from β-C to the active sites is relatively easy. The β-C, α-C and N atoms of tri-hydrogen-quinoline and the Mo, Ni and S atoms on the active sites form a stable hexa-cycle. However, the reaction energy and activation energy of the C–N cleaved on THQ-1 (the second step of E_2_) are both higher than that of MHI. As mentioned above, for an elemental step in heterogeneous catalysis, the reaction energy and the activation energy are largely determined by the combination ability between the reactants (also including the transition state, intermediate and product) and the active sites. For DHI, the product of the C–N cleavage (i.e., o-vinyl aniline) is a planar configuration because all of the carbon atoms are sp^2^ hybridization (as shown in Table 8). The C=C double bonds and the –NH groups are in the same plane and suitable for adsorption on the active sites, whereas the product of THQ-1 (i.e., o-propenyl aniline) has an sp^3^ hybridized γ-carbon atom, leading to a non-coplanar configuration between the C=C double bonds and the –NH radical groups. The synchronous adsorption of the C=C bonds and –NH groups on the active sites will cause severe deformation of the nitrogen compounds. Meanwhile, the sp^3^ γ-C obstructs the conjugated system between the β-C and the aromatic ring, which will further elevate the system energy at the transition state of the C–N bond cleavage for THQ-1.

When the THQ-1 is fully hydrogen saturated to the DHQ, the rigidity of the aromatic ring relaxes, and the deformation of the naphthenic ring is easier (Table 9). The C‒N bond cleavage of DHQ has a lower activation energy than that of THQ-1, and it can be considered the dominant reaction pathway of the C‒N bond cleavage in the HDN network of quinoline. The calculation results can also explain the experiments in which the content of OPA was very low in the product of quinoline HDN under moderate temperature, as in the experiments mentioned above. They can also explain the experiments in other reports in which, as the temperature increased, the selectivity of OPA increased because the activation energy of THQ-1 to OPA was higher than that of DHQ [44].

### 2.2. Substituent Effects of the Heterocycle on the HDN Behavior 

#### 2.2.1. Substituent Effects on the Indole HDN

The HDN performances of 2-methyl-indole (2-M-IND) and 3-methyl-indole (3-M-IND) on the Ni-Mo/Al_2_O_3_ catalyst under various LHSVs are listed in Table 10. Compared with the HDN data on indole, as shown in Table 3, the conversion rate of 2-M-IND and 3-M-IND were both lower than that of indole under the same reaction condition. Furthermore, the contents of dihydro-2-methyl-indole (DH-2-M-IND) and dihydro-3-methyl-indole (DH-3-M-IND) were lower than that of DHI in the indole HDN product, indicating that the methyl group on α-C and β-C can both limit the hydrogenation saturation of the indole nitrogen heterocycle. 

#### 2.2.2. Theoretical Explanation

According to Table 3, the Ni-S-edge is the dominant active site for the hydrogenation saturation of the indole heterocycle. The conversions of the methyl-substituted indole on the Ni-S-edge are listed in Table 11. The optimized adsorption morphology of indole is vertical adsorption, with its C=C double bonds contacting with two Ni atoms of the Ni-S-edge simultaneously. The methyl group on the α-C or β-C will not evidently influence the indole adsorption because the entire nitrogen compound is still a planar structure, and it can still be settled in the narrow active site created by the H_2_S desorption. However, when the hydrogenation process begins, one sp^2^ carbon on the nitrogen heterocycle converts to sp^3^ hybridization. The methyl-substituted nitrogen compound is not a planar structure, and the spatial requirement for the adsorption increases notably. The narrow room of the active sites on the Ni-S-edge cannot hold the stereo nitrogen compound well, and the system energy markedly increases at the transition state and the final state of the hydrogen transfer. It can be concluded that the substituent group on the nitrogen heterocycle increases the stereo special requirements during the hydrogenation of the corresponding aromatic carbon, and this is the prime reason that the conversion rate of 2-methyl-indole and 3- methyl-indole is lower than that of indole at the same HDN reaction condition.

#### 2.2.3. Substituent Effects on the Quinoline Self-HDN Behavior on Ni-Mo-S/γ-Al_2_O_3_


The HDN performances of 2-methyl-quinoline (2-M-QL) and 3-methyl-quinoline (3-M-Ql) on the Ni-Mo-S/Al_2_O_3_ catalyst under various LHSVs are listed in Table 12. Compared with the HDN data of quinoline in Table 2, it can be found that the HDN rate of 2-M-QL was also quite similar to that of QL under the same reaction condition, and the product distribution of these two nitrogen compounds was also much alike, indicating that the methyl group on the α-C did not obviously change the self-HDN behavior of 2-M-QL. However, the HDN rate of 3-M-QL was notably lower than that of QL under the same reaction conditions. In the HDN product of 3-M-QL, it can be founded that the content of 3-methyl-decahydroquinoline (3-M-DHQ) was much higher than that of DHQ in the quinoline HDN products, indicating that the methyl group on the β-C hindered the C‒N bond cleavage of 3-M-DHQ. 

#### 2.2.4. Theoretical Explanation

According to the calculations in Table 7, it can be predicted that the α-C and β-C atoms must push aside the –SH group on the Ni-Mo-edge before the C‒N bond can completely break. During this process, the C=C double bonds belonging to the nitrogen heterocycle have to rotate approximately 90°, with the β-C as the axle (from parallel to the Ni-Mo-edge to the perpendicular). The C‒N bond cleavage of the methyl-substituted quinoline on the Ni-Mo-edge is listed in Table 13. When the methyl group is located on the α-C atom of the nitrogen heterocycle, this methyl group does not contact with the –SH, and the resistance of the rotation does not evidently increase; therefore, the activation energies of the C‒N bond cleavage of DHQ and 2-M-DHQ on the Ni-Mo-edge are both approximately 170–175 kJ·mol^−1^. When the methyl group locates on the β-C atom of the nitrogen heterocycle, this methyl group rotates as an additional branch of the β-C atom, and it has to press the –SH groups together with the β-C atom at the transition state. The activation energy of this step of 3-M-DHQ is 201.08 kJ·mol^−1^, approximately 30 kJ·mol^−1^ larger than that of DHQ. This could be the reason that 2-M-QL and QL have similar HDN behavior, whereas the HDN rate of 3-M-QL is notably lower than that of QL.

#### 2.2.5. Substituent Effects on Quinoline HDN Competitive Ability

The basic nitrogen compounds have stronger adsorptivity because of the lone pair electrons on the nitrogen atom. The methyl group on the nitrogen heterocycle might influence the contact between the lone pair of electrons of the N atom and the exposed metal atom on the active site and change the HDN competitive ability of 2-M-QL and 3-M-QL. The HDN product distribution of the mixed 2-M-QL and QL reactants catalyzed on Ni-Mo-S/γ-Al_2_O_3_ are listed in Table 14. Compared with Table 2 and Table 12, it can be founded that the HDN rate of 2-M-QL in the mixed reactant was notably lower than that of 2-M-QL itself, whereas the HDN behavior of QL was not evidently influenced by the 2-M-QL, indicating the competitive ability of 2-M-QL was weaker than that of QL. The HDN productions of the mixed 3-M-QL and QL are listed in Table 15. It can be found that 3-M-QL and QL basically kept their respective HDN behavior, and they did not interfere with each other, indicating that the competitive abilities of 3-M-QL and QL were quite similar.

#### 2.2.6. Theoretical Explanation

The molecular orbital and the adsorption of QL, 2-M-QL and 3-M-QL on the Ni-Mo-edge are listed in Table 16. It can be found that the methyl groups on neither 2-M-QL nor 3-M-QL can obviously change the molecular orbital of the lone pair of electrons. However, the methyl group on 2-M-QL is in the same extension direction with the lone pair of electrons, and it will weaken the adsorbing N–Ni bond. On the other hand, the methyl group on 3-M-QL is in the opposite direction against the lone pair of electrons, and the adsorbing N–Ni bond of 3-M-QL is not influenced. Therefore, the adsorption energy values of QL and 3-M-L are both approximately −150 kJ·mol^−1^, and the one of 2-M-QL is merely approximately −120 kJ·mol^−1^; this might be the reason why the methyl group on 2-M-QL weakens the HDN competitive ability, whereas the ones on 3-M-QL do not.

## 3. Materials and Methods

### 3.1. Experimental

#### 3.1.1. Preparation of the NiMo/γ-Al_2_O_3_ Catalysts

A quantity of 60.0 g pseudoboehmite(Ying Lang Chemical, Shandong, China) was mixed evenly with 70.0 g deionized water, 2.0 g citric acid(Innochem, Beijing, China), 2.0 g acetic acid(Aldrich, Burlington, MA, USA) and 1.0 g sesbania powder(Zhejiang Lvzhou Bio-tec. Co., Ltd., Hangzhou, China). The mixture was extruded into strips with a 1.7 nm diameter. The strips were dried in the air for 48.0 h and at 393 K for 4.0 h. Then, the strips were calcined at 873 K for 4.0 h and then shaped into 0.6–0.8 mm to obtain the γ-Al_2_O_3_ supports. The NiMo/γ-Al_2_O_3_ catalyst was prepared by incipient wetness impregnation: 30.0 g γ-Al_2_O_3_ supports were loaded with 5.4 g ammonium molybdate tetrahydrate(Aldrich, USA) and 2.4 nickelnitrate(Innochem, China), which were dissolved in 35.0 g ammonia water with a pH ranging from 10 to 11. Finally, the catalyst was dried and calcined at 823 K for 6 h.

#### 3.1.2. HDN Evaluation and Analysis

The HDN reactions were carried out in a fixed-bed reactor with an 18 mm inner diameter and a 5000 mm length loaded with 6.0 g of catalyst diluted to 20.0 mL with quartz sand. The catalysts need presulfurization with a CS_2_ cyclohexane solution with a CS_2_ concentration of 4.0 vol % at 623 K and 4.0 MPa for 5.0 h, with a weight hourly space velocity (WHSV) of 10 h^−1^ and an H_2_/oil ratio of 300 (*v/v*). The HDN reactions were performed with different WHSVs in the range from 2.5 to 20.0 h^−1^. The H_2_/oil ratio was fixed at 400 (*v/v*), and the temperature was 573 K. The HDN reaction atmosphere was 4.0 MPa H_2_ with a 1.0% partial pressure of H_2_S. 

The liquid products were collected after a stabilization period of 24.0 h and analyzed on an Agilent 4890D gas chromatograph with a photoionization detector. The chromatographic conditions are set as follows: the initial column temperature was 40.0 ℃. The temperature increase in the column was 2.0 ℃/min, and the end temperature was 320 ℃. The mass fraction of each product was measured by the peak area. 

To further identify each compound in the liquid products, a Finnigan TRACE gas chromatography/mass spectrometry (GC/MS) system consisting of a TRACE Ultra gas chromatograph capillary column and an HP 5973 MS detector was used to analyze the collected products. The peak position for each product is listed in the Appendix A.

### 3.2. Theoretical Calculation

#### 3.2.1. Modeling 

The framework of the model Ni-Mo-S nanocluster was based on scanning tunneling microscopy (STM) characterization [45,46], and the edge structure was based on the periodic structure proposed in other studies [32,47,48]. On the Ni-Mo-edge, the nickel coverage was 50%. The Ni atom was square-planar coordination with four S atoms, and the normal direction of the square plane was exposed without S atoms. The calculation model was established, as shown in Figure 1 [15]. 

#### 3.2.2. Computational Methods

The calculations were performed using the DMol^3^ code with numerical atomic functions [49,50]. The calculation software is integrated in Materials Studio 8.0 designed by Accelrys Ltd., San Diego, CA, USA. This calculation strategy balances the calculation speed and accuracy, which is suitable for the preliminary quantitative calculation of the nonperiodic structures. To analyze the transition state, the open shell mode was used to treat the electron spin. To reduce the energy error caused by the basis set, the BSSE correction was used in the calculation of the adsorption. The complete linear synchronous transit (LST) and quadratic synchronous transit (QST) methods were used to determine the transition state, and the nudged elastic band (NEB) method was used to confirm the transition state. Other calculation details and parameters are listed in Table 17.

## 4. Conclusions

Combining the HDN experiments and the theoretical calculations, some HDN behavioral characteristics of indole and quinoline were explained, and the methyl substituent effects of indole and quinoline were also studied. One common point b indole and quinoline HDN was the rapid hydrogenation saturation of the nitrogen heterocycle. The theoretical calculations predicted that this conversion was more likely to take place on the Ni-S-edge of the Ni-Mo-S nanocluster. Another common point was that the C‒N bond cleavages of indole and quinoline were both via the *E_2_* pathway on the Ni-Mo-edge. The S atom on the Ni-Mo-edge first received the β-hydrogen of the nitrogen heterocycle, and then the C‒N bond began to break with the rotation of C=C double bonds. The notable difference between indole and quinoline was that the primary hydrogenation product of indole could directly break the C‒N bond, whereas the primary hydrogenation product of quinoline can break the C‒N bond only after the full hydrogenation saturation to decahydroquinoline. The reason is that there is only the sp^2^ hybrid α-C and β-C on the nitrogen heterocycle of indole, whereas there is an additional γ-C atom on the nitrogen heterocycle of quinoline. When the C‒N bond breaks, the ‒NH and C=C groups of DHI can be coplanar, whereas those of THQ cannot because of the sp^3^ hybrid γ-C.

Whenever the methyl group locates on the α-C or the β-C atom of indole, the hydrogenation ability of the nitrogen heterocycle will be evidently weakened. The methyl group increases the space requirement of the sp^3^ carbon, and the impaction of the C=C groups on the CUS of the Ni-S-edge cannot provide enough space. When the methyl groups are located on the α-C of quinoline, the self-HDN behavior of 2-methyl-quinoline is similar to quinoline, whereas the competitive HDN ability of 2-methyl-quinoline in the homologs is evidently weakened. The methyl group on the α-C hinders the contact between the N atom of 2-methyl-quinoline and the exposed metal atom of the CUS sites. When the methyl group locates on the β-C of quinoline, the C‒N bond cleavage of 3-methyl-quinoline becomes more difficult because the methyl group on the β-C increases the steric hindrance of the C=C group during the process. However, the competitive HDN ability of 3-methyl-quinoline is not evidently influenced because the methyl group on the β-C does not evidently hinder the adsorption of 3-methyl-quinoline on the active sites.

## Figures and Tables

**Figure 1 ijms-24-03044-f001:**
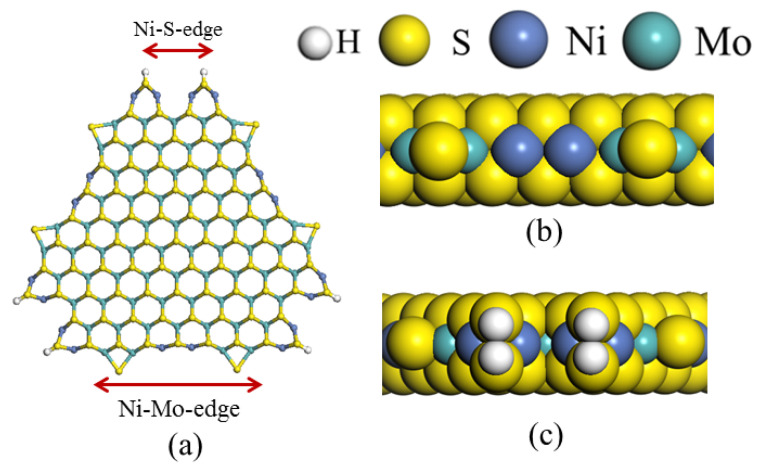
Configuration of the model Ni-Mo-S nanocluster: (**a**) schematic of the nanocluster; (**b**) side view of the Ni-Mo-edge; (**c**) side view of the Ni-S-edge.

**Table 1 ijms-24-03044-t001:** HDN products of indole and quinoline.

Indole			Quinoline		
Product	Formula	Structural	Product	Formula	Structural
Ethyl-cyclohexane (ECH)	C_8_H_16_	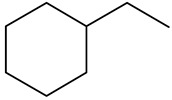	Propyl-cyclohexane (PCH)	C_9_H_18_	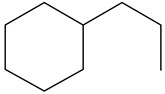
Ethyl-benzene (EB)	C_8_H_10_	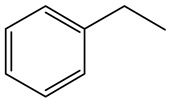	Propyl-benzene (PB)	C_9_H_12_	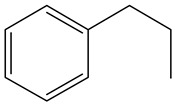
Ethyl-cyclohexene (ECHE)	C_8_H_14_	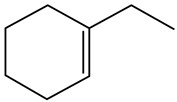	Propyl-cyclohexene (PCHE)	C_9_H_16_	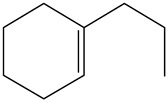
o-Ethylaniline (OEA)	C_8_NH_15_	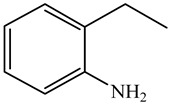	Decahydroquinoline (DHQ)	C_9_NH_17_	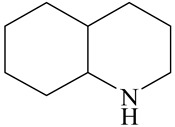
Indoline (DHI)	C_8_NH_9_	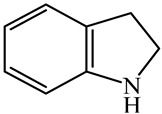	o-Propylaniline (OPA)	C_9_NH_13_	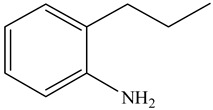
Indole (IND)	C_8_NH_7_	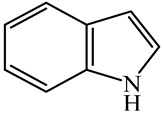	5,6,7,8-Tetrahydroquinoline (THQ-5)	C_9_NH_11_	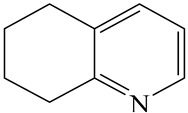
			Quinoline (QL)	C_9_NH_7_	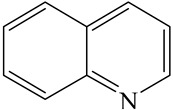
			1,2,3,4-Tetrahydroquinoline (THQ-1)	C_9_NH_11_	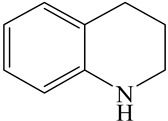

**Table 2 ijms-24-03044-t002:** Product distribution of the indole and quinoline HDN on Ni-Mo-S/γ-Al_2_O_3_.

Reactant		Relative Concentration, m/m%							
**Indole**	**LHSV/h^−1^**	**IND**	**DHI**	**OEA**	**ECHE**	**EB**	**ECH**		
	20.0	60.92	20.96	7.42	2.67	2.54	5.49		
	10.0	36.52	12.14	18.24	4.84	2.86	25.16		
	5.0	18.19	7.48	18.94	3.92	1.57	49.95		
	2.5	7.03	1.92	3.81	1.02	1.34	85.39		
**Quinoline**	**LHSV/h^−1^**	**THQ-1**	**QL**	**THQ-5**	**OPA**	**DHQ**	**PCHE**	**PB**	**PCH**
	20.0	79.77	1.92	8.20	0.22	6.16	1.10	0.29	2.34
	10.0	64.89	1.49	11.83	1.10	10.99	2.45	0.52	7.25
	5.0	36.24	0.74	15.86	1.65	15.64	3.67	2.12	24.22
	2.5	21.46	0.39	15.30	1.82	11.70	2.77	1.82	43.74

**Table 3 ijms-24-03044-t003:** Conversion from indole to indoline on the Ni-S-edge and Ni-Mo-edge.

**Position**	**Step**	**Indole ** **Adsorption**	**TS-1**	**Hydrogen** **Activation**	**TS-2**
Ni-S-edge	Morphology	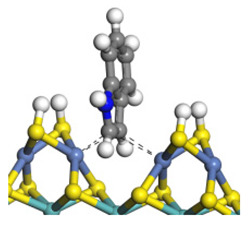	-	-	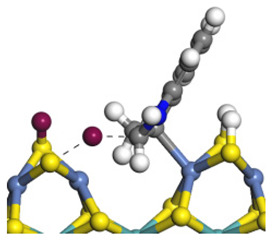
ΔE/kJ·mol^−1^	−124.18			+128.27
Ni-Mo-edge	Morphology	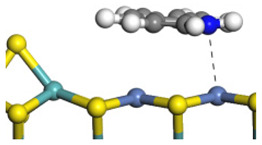	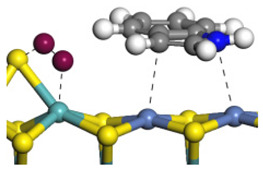	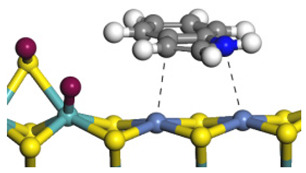	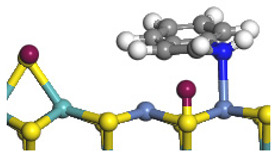
ΔE/kJ·mol^−1^	−121.63	+145.09	−108.42	+129.65
**Position**	**Step**	**Monohydrogen-** **Indole**	**TS-3**	**DHI**	**Desorption**
Ni-S-edge	Morphology	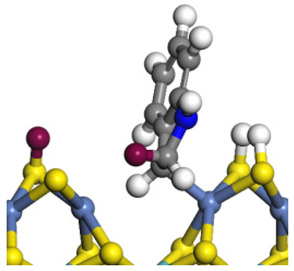	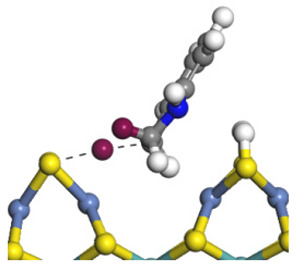	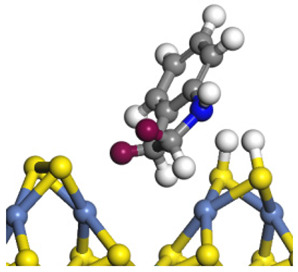	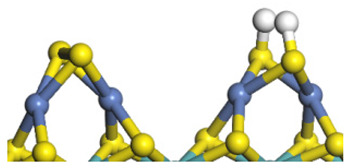
ΔE/kJ·mol^−1^	−96.20	+108.75	−202.39	+163.85
Ni-Mo-edge	Morphology	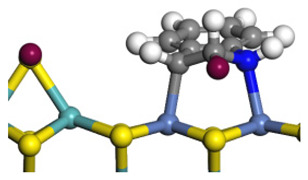	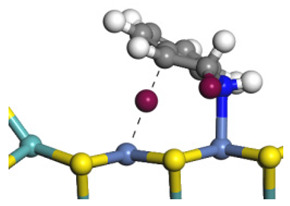	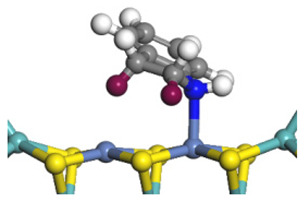	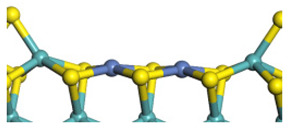
ΔE/kJ·mol^−1^	−133.33	+101.24	−220.00	
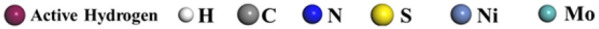

**Table 4 ijms-24-03044-t004:** Conversion from quinoline to tetrahydroquinoline on the Ni-S-edge and Ni-Mo-edge.

Position	Step	QuinolineAdsorption	TS-1	HydrogenActivation	TS-2	Monohydrogen-Quinoline	TS-3	Dihydrogen-Quinoline
Ni-S-edge	Morphology	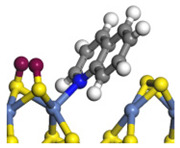	-	-	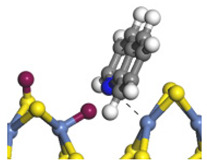	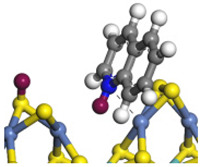	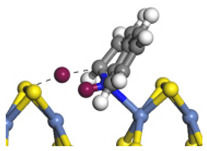	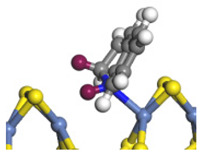
ΔE/kJ·mol^−1^	−169.22			+140.22	−109.88	+114.82	−87.56
Ni-Mo-edge	Morphology	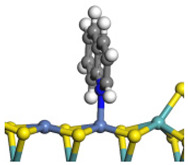	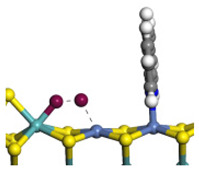	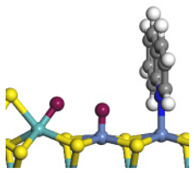	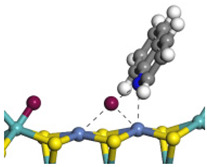	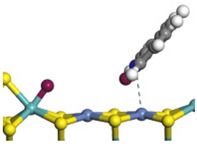	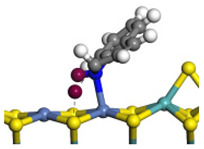	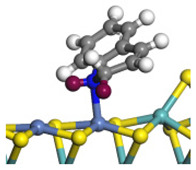
ΔE/kJ·mol^−1^	−146.34	+90.67	−85.79	+115.98	−137.52	+64.83	−82.22
	**Step**	**TS-4**	**Hydrogen** **Activation**	**TS-5**	**Tri-** **hydrogen** **-quinoline**	**TS-6**	**THQ-1**	**Desorption**
Ni-S-edge	Morphology	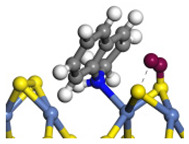	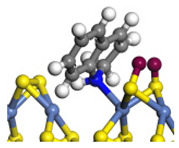	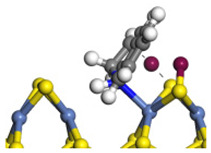	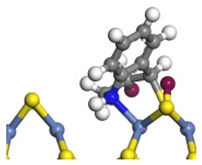	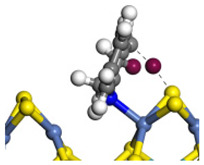	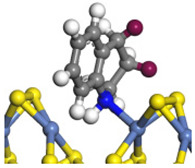	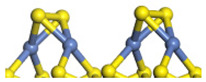
ΔE/kJ·mol^−1^	+127.12	−186.74	+102.68	−110.58	+85.99	−143.26	+163.85
Ni-Mo-edge	Morphology	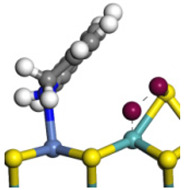	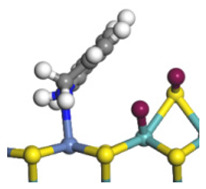	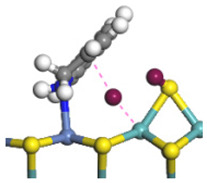	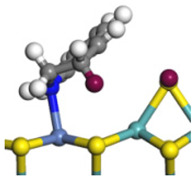	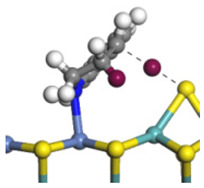	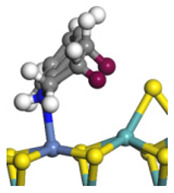	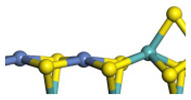
ΔE/kJ·mol^−1^	+144.33	−100.79	+95.55	−81.09	+76.77	−198.53	
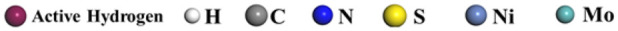

**Table 5 ijms-24-03044-t005:** The S_N_2 and E_2_ routes of the C–N bond cleavage of DHI on the Ni-Mo-edge.

Reaction Pathway	S_N_2	E_2_ Step1	E_2_ Step2
Reaction equation	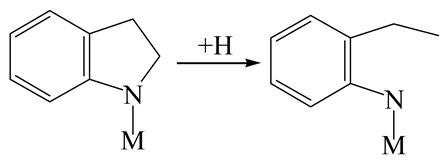	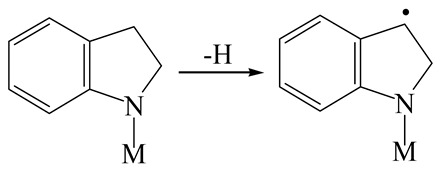	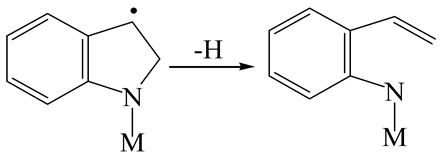
Pre-hydrogenolysis	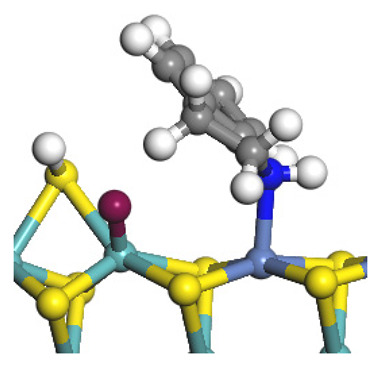	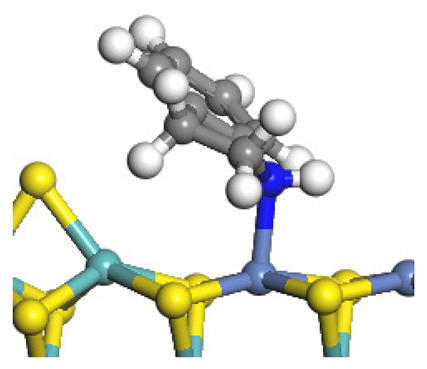	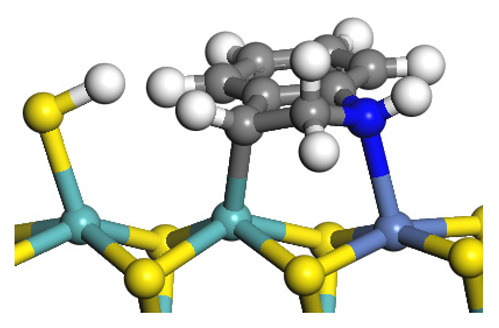
Transition state	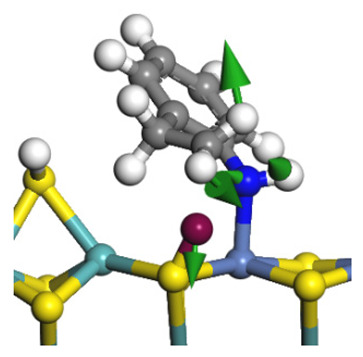	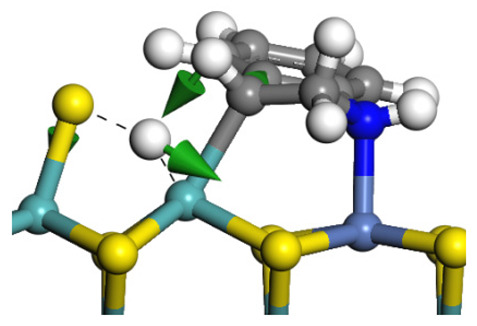	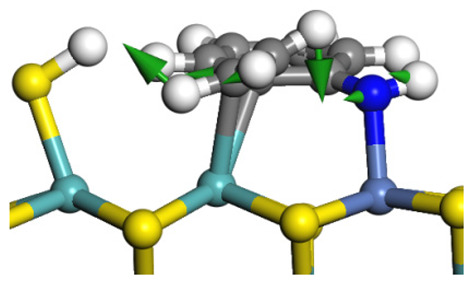
Post-hydrogenolysis	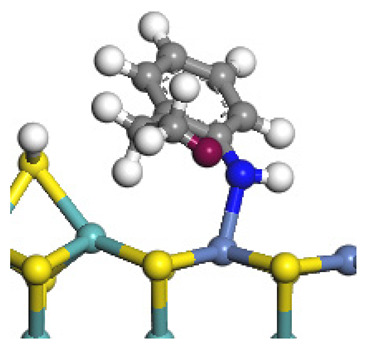	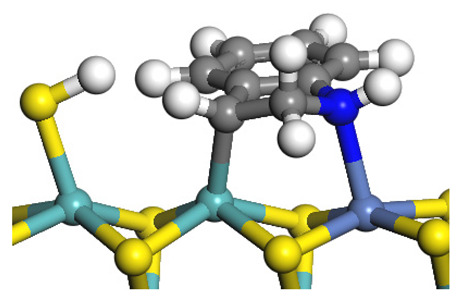	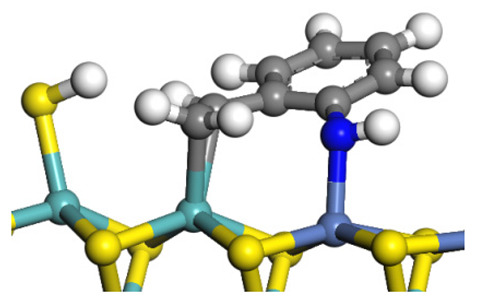
Activation energy/kJ·mol^−1^	+269.39	+136.93	+132.47
Reaction energy/kJ·mol^−1^	−57.87	+56.75	+59.02
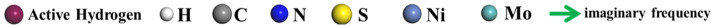

**Table 6 ijms-24-03044-t006:** Molecular orbitals and adsorption of HDI and its intermediates of the C–N bond cleavage.

Nitrogen Compounds	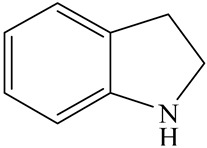 DHI	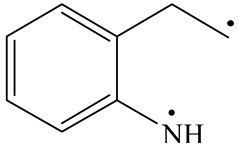 C–N Bond Cleavage
Bonding Energy Difference/kJ·mol^−1^	281.84
Molecular orbitals participating in adsorption	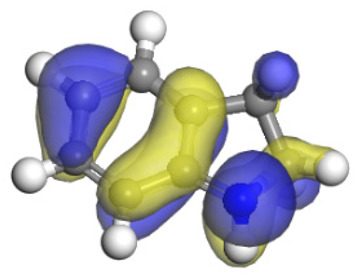	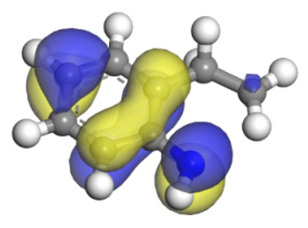	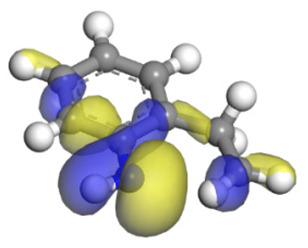
MO Eigenvalue/eV	−4.56	−4.51	−5.76
Adsorptionmorphology	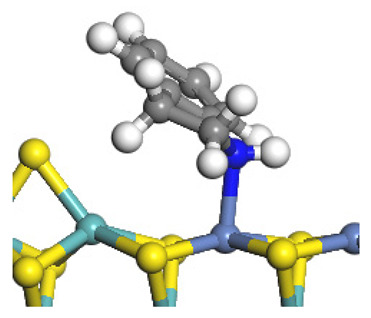	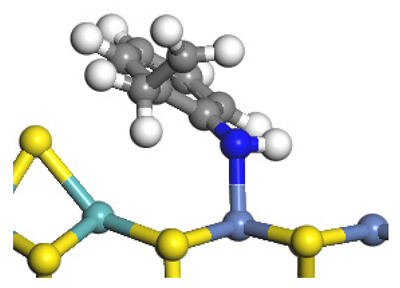
Adsorption energy/kJ·mol^−1^	−156.88	−258.59
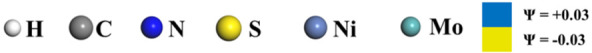

**Table 7 ijms-24-03044-t007:** The C–N bond cleavage of THQ-1 on the Ni-Mo-edge.

Reaction	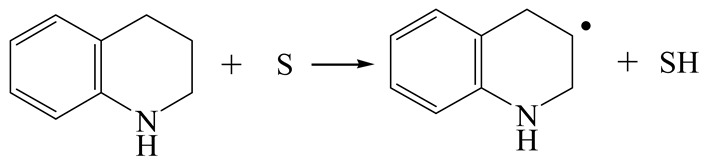	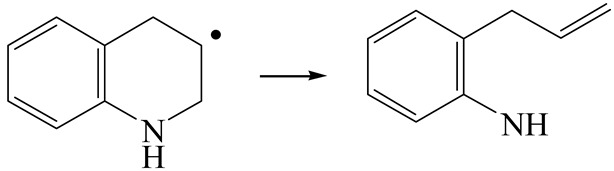
Initial state	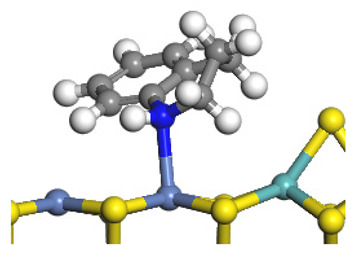	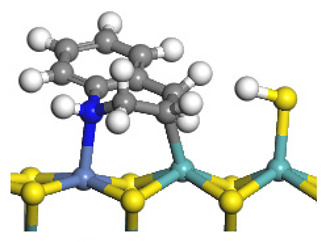
Transition state	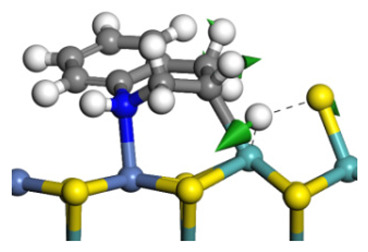	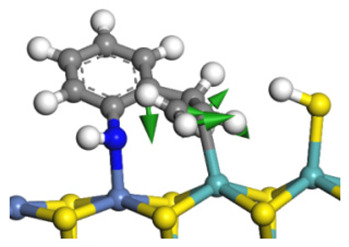
Final state	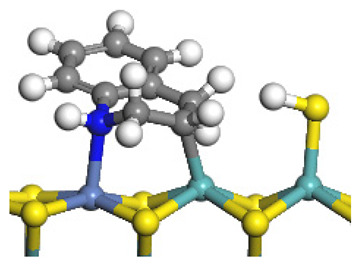	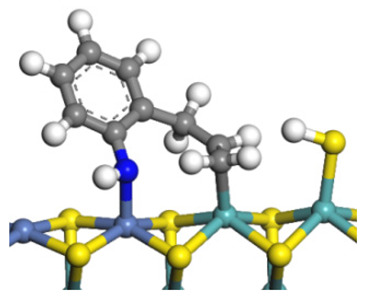
Activation energy/kJ·mol^−1^	+142.37	+208.81
Reaction energy/kJ·mol^−1^	+75.97	+95.02
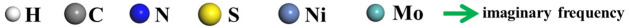

**Table 8 ijms-24-03044-t008:** Molecular orbitals of o-vinyl aniline and o-propenyl aniline.

NitrogenCompounds	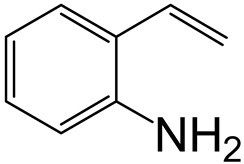	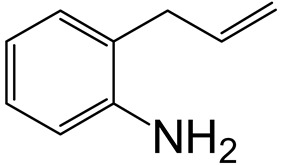
Molecular orbitalsparticipate in adsorption	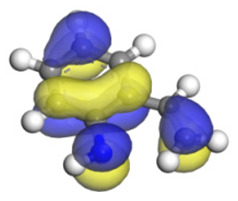	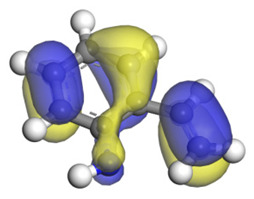	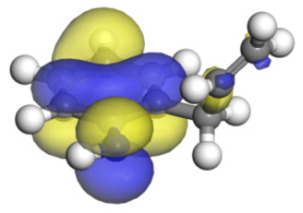	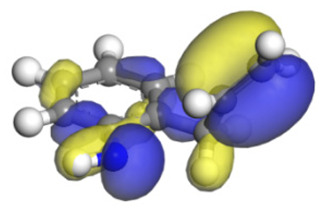
MO Eigenvalue/eV	−4.84	−5.72	−4.91	−5.90
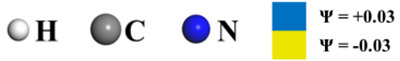

**Table 9 ijms-24-03044-t009:** The C‒N bond cleavage of DHQ on the Ni-Mo-edge.

Reaction	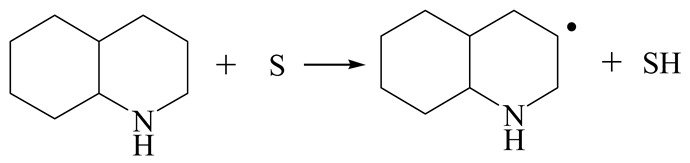	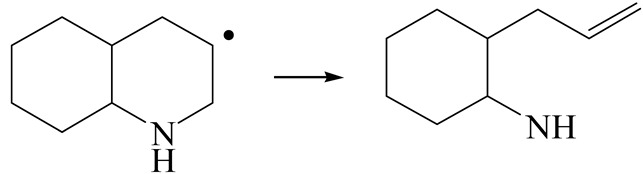
Initial state	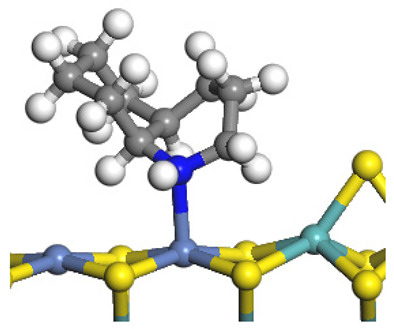	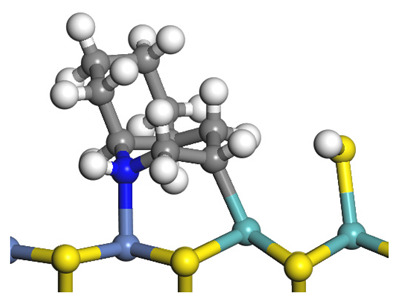
Transition state	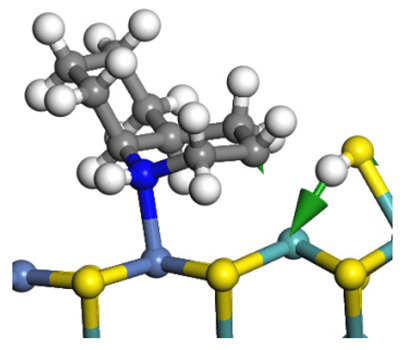	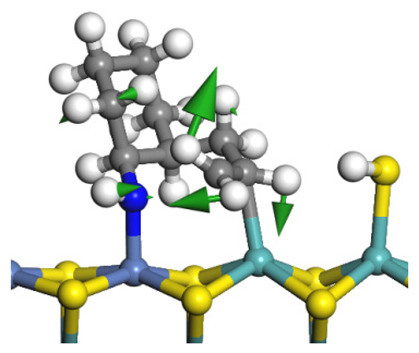
Final state	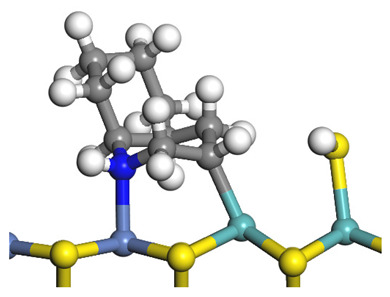	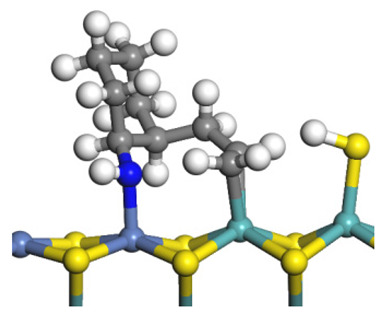
Activation energy/kJ·mol^−1^	+130.25	+172.85
Reaction energy/kJ·mol^−1^	+81.54	+80.44
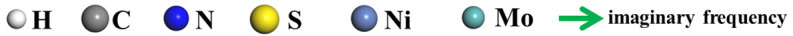

**Table 10 ijms-24-03044-t010:** HDN performances of 2-methyl-indole and 3-methyl-indole on Ni-Mo-S/γ-Al_2_O_3_.

Reactant	Relative Concentration of HDN Products, m/m%						
**2-Methyl** **-indole**	**LHSV/h^−1^**	** 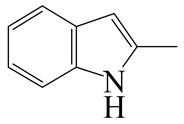 **	** 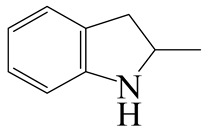 **	** 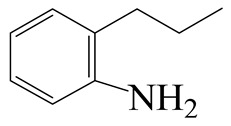 **	** 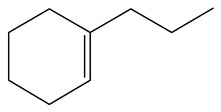 **	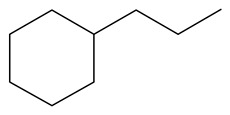	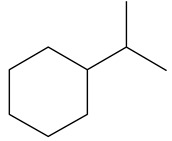
		**2-M-IND**	**DH-2-M-IND**	**OPA**	**PCHE**	**PCH**	**i-PCH**
	20.0	68.96	10.84	7.42	1.91	14.17	0.59
	10.0	46.29	6.56	8.24	1.52	33.13	1.19
	5.0	27.40	2.09	4.33	2.09	52.95	2.81
	2.5	13.73	1.73	2.42	1.95	75.86	3.71
**3-Methyl** **-indole**	**LHSV/h^−1^**	**3-M-IND**	**DH-3-M-IND**	**i-OPEA**	**i-OPA**	**i-PCHE**	**i-PCH**
		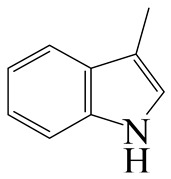	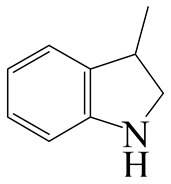	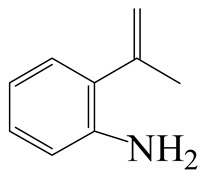	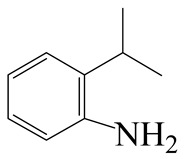	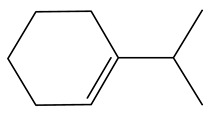	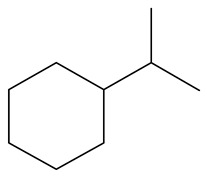
	20.0	78.84	4.93	1.21	3.53	2.19	9.30
	10.0	59.14	4.82	1.24	5.76	2.17	26.87
	5.0	40.56	1.85	1.57	5.81	2.31	47.90
	2.5	22.11	0.66	1.58	3.73	1.43	70.49

**Table 11 ijms-24-03044-t011:** Hydrogenation of the methyl-substituted indole heterocycle on the Ni-S-edge.

Reaction	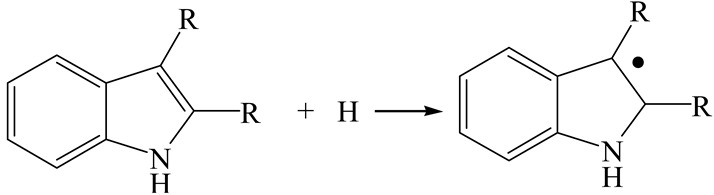
Reactant	Indole	2-Methyl-Indole	3-Methyl-Indole
Adsorption morphology	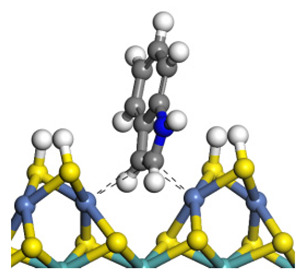	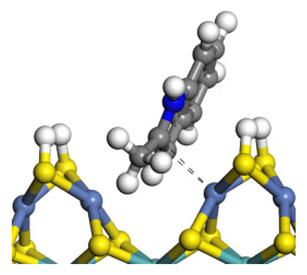	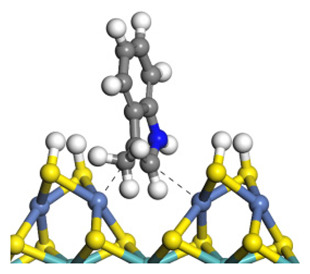
Adsorption energy/kJ·mol^−1^	−124.18	−131.32	−117.66
Transition state	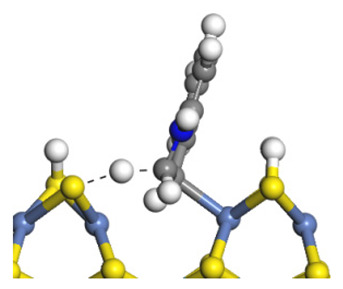	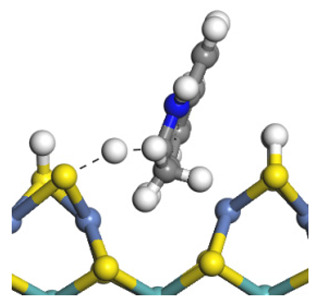	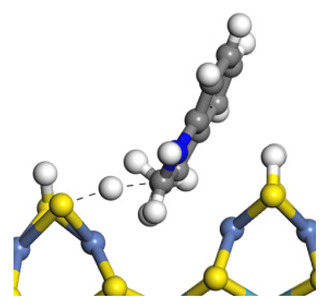
Activation energy/kJ·mol^−1^	+128.27	+151.35	+156.48
Final state	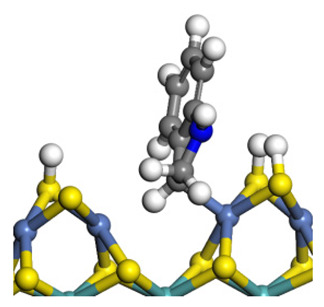	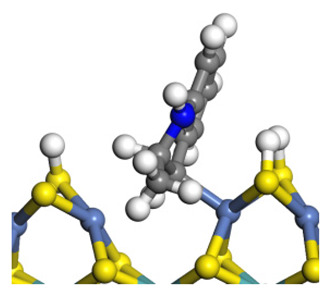	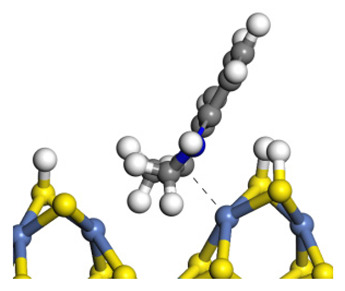
Reaction energy/kJ·mol^−1^	+32.29	+60.74	+72.57
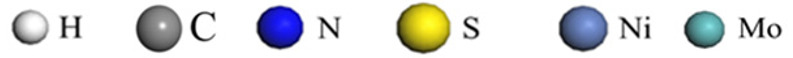

**Table 12 ijms-24-03044-t012:** Products of 2-methyl-quinoline and 3-methyl-quinoline on Ni-Mo-S/γ-Al_2_O_3_.

Relative Concentration, m/m%								
**2-Methyl** **-quinoline** **LHSV/h^−1^**	**2-M-** **THQ-1**	**OBA**	**2-M-QL**	**2-M-** **THQ-5**	**2-M-DHQ**	**1-BCHE**	**2-BCHE**	**BCH**
	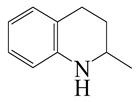	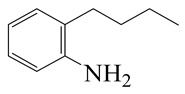	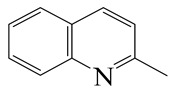	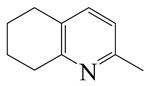	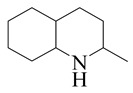	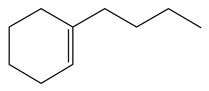	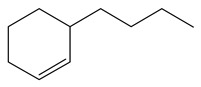	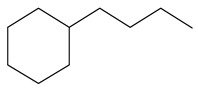
20.0	72.57	2.04	4.17	9.03	7.21	1.16	0.82	3.06
10.0	57.05	4.33	4.16	12.49	11.30	2.17	1.63	6.87
5.0	35.94	5.47	3.27	14.20	12.03	1.71	0.13	27.25
2.5	22.49	1.16	2.20	14.75	11.24	1.22	1.29	45.65
**3-Methyl** **-quinoline** **LHSV/h^−1^**	**3-M-** **THQ-1**	**3-M-QL**	**3-M-** **THQ-5**	**i-OBA**	**3-M-DHQ**	**1-i-BCHE**	**2-i-BCHE**	**i-BCH**
	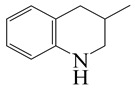	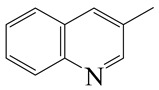	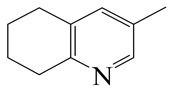	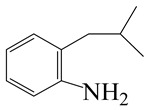	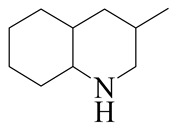	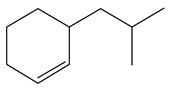	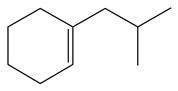	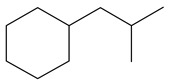
20.0	72.73	4.88	11.27	0.81	8.16	0.58	0.11	1.46
10.0	59.10	3.95	14.49	1.73	13.55	1.24	0.23	5.71
5.0	36.69	3.84	17.13	2.36	19.91	2.29	0.40	15.37
2.5	24.88	2.50	13.14	2.39	25.22	1.26	1.88	28.73

**Table 13 ijms-24-03044-t013:** The **C–N** bond cleavage of methyl-substituted quinoline on the Ni-Mo-edge.

Reaction	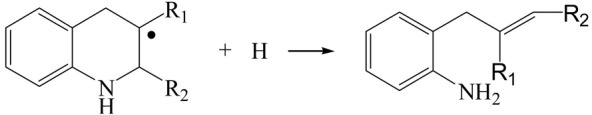
Reactant	DHQ	2-M-DHQ	3-M-DHQ
Initial state	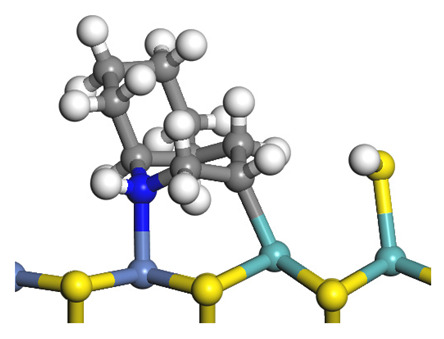	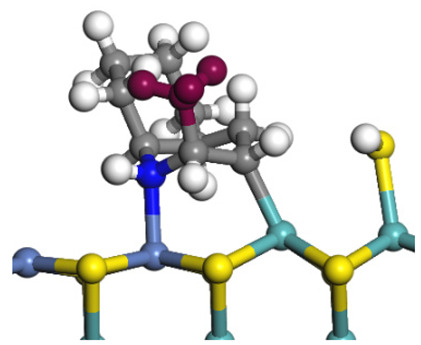	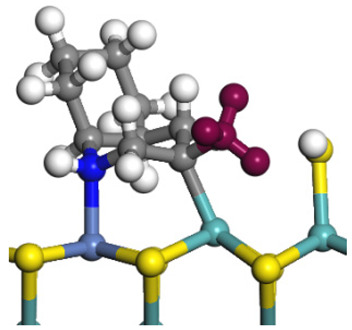
Transition state	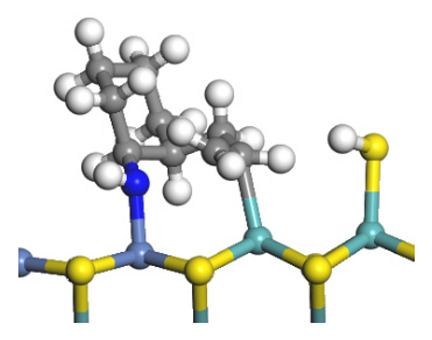	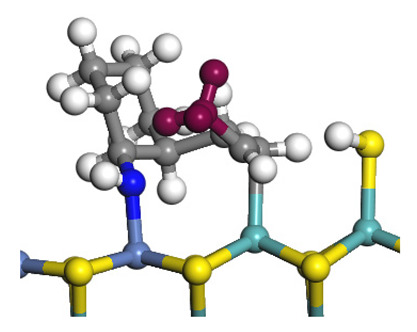	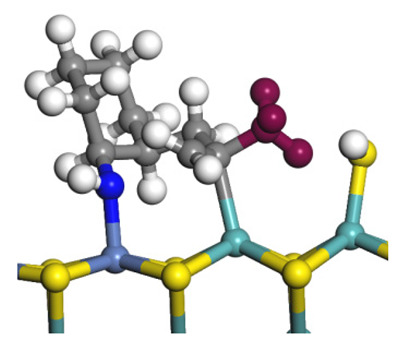
Activation energy/kJ·mol^−1^	+172.85	+174.59	+201.08
Final state	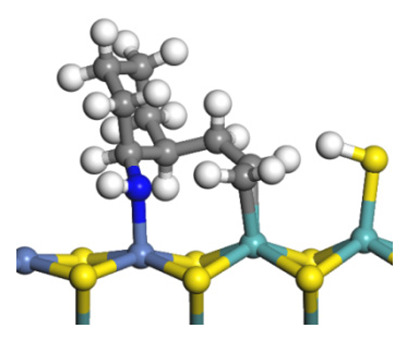	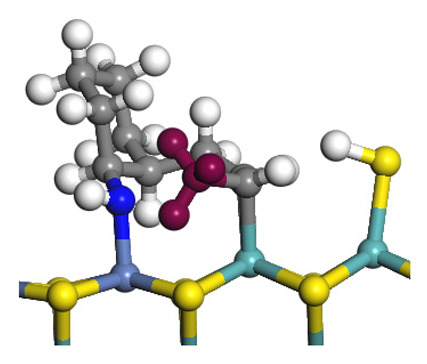	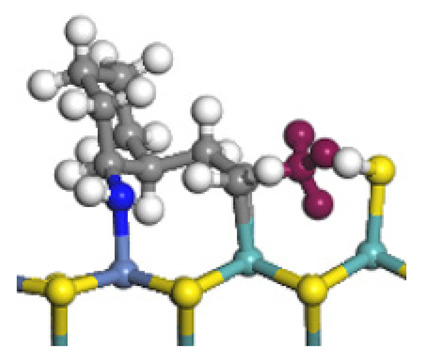
Reaction energy/kJ·mol^−1^	+80.44	+86.71	+82.93
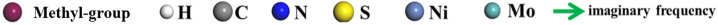

**Table 14 ijms-24-03044-t014:** Product distribution of the 2-M-QL and QL mixed feedstock on Ni-Mo-S/γ-Al_2_O_3_.

Relative Concentration, m/m%									
**2-M-QL**	**LHSV/h^−1^**	**2-M-** **THQ-1**	**OBA**	**2-M-Q**	**2-M-** **THQ-5**	**2-M-DHQ**	**1-BCHE**	**2-BCHE**	**BCH**
	20.0	57.24	0.62	27.13	9.51	4.73	0.11	0.24	0.42
	10.0	48.86	2.61	25.62	12.59	9.31	0.14	0.17	0.70
	5.0	44.55	2.38	13.2	15.13	17.43	0.27	0.25	6.79
	2.5	26.85	4.96	7.27	12.26	25.5	1.14	0.41	21.61
**QL**	**LHSV/h^−1^**	**THQ-1**	**Q**	**THQ-5**	**DHQ**	**PB**	**PCHE**	**PCH**	
	20.0	74.63	7.07	9.64	5.52	1.19	0.29	1.66	
	10.0	61.79	6.12	13.7	12.2	1.83	0.45	3.91	
	5.0	47.31	3.3	19.24	11.27	3.41	1.33	14.14	
	2.5	24.28	2.32	24.76	7.11	2.15	1.13	38.25	

**Table 15 ijms-24-03044-t015:** Product distribution of the 3-M-QL and QL mixed feedstock on Ni-Mo-S/γ-Al_2_O_3_.

Relative Concentration, m/m%									
**3-M-QL**	**LHSV/h^−1^**	**3-M-** **THQ-1**	**3-M** **-QL**	**3-M-** **THQ-5**	**i-** **OBA**	**3-M** **-DHQ**	**1-i-** **BCHE**	**2-i-** **BCHE**	**i-** **BCH**
	20.0	60.22	13.54	12.19	5.54	4.78	0.21	0.09	3.43
	10.0	56.34	11.67	12.01	3.82	11.07	0.25	0.1	4.74
	5.0	44.59	8.22	16.17	3.2	12.95	1.32	0.76	12.79
	2.5	29.24	4.42	16.49	3.02	18.34	2.00	1.12	25.37
**QL**	**LHSV/h^−1^**	**THQ-1**	**Q**	**THQ-5**	**DHQ**	**PB**	**PCHE**	**PCH**	
	20.0	76.13	12.51	1.36	7.59	0	0.53	1.88	
	10.0	71.51	9.06	1.49	11.08	0	1.50	5.36	
	5.0	54.48	5.13	6.97	13.2	1.29	3.11	15.82	
	2.5	35.46	2.7	10.49	8.79	1.54	1.29	39.73	

**Table 16 ijms-24-03044-t016:** Molecular orbital and adsorption data of QL, 2-M-QL and 3-M-QL on the Ni-Mo-edge.

Nitrogen Compound	Molecular Orbital of Lone Pair Electrons	Adsorption Morphology	N-NiBond Length/Å	Adsorption Energy/kJ·mol^−1^
Quinoline	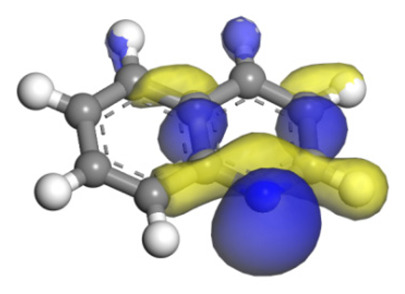	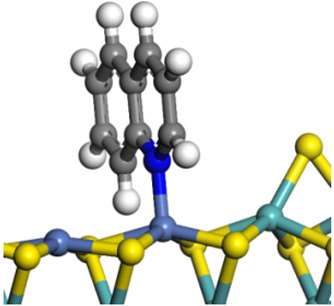	2.11	−146.34
2-Mythel-quinoline	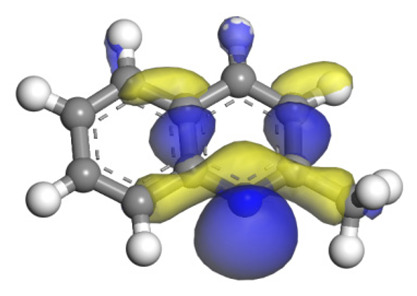	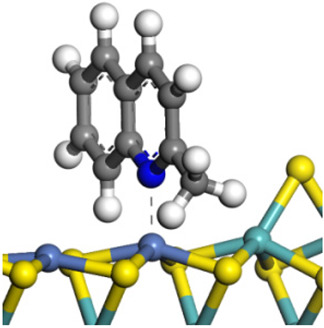	2.19	−118.87
3-Mythel-quinoline	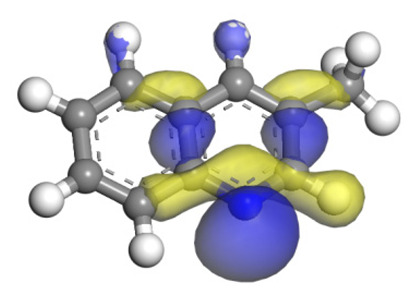	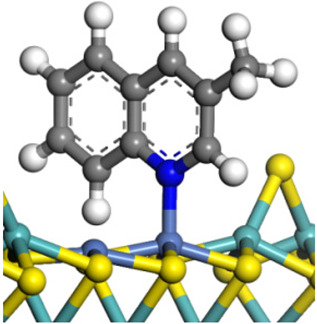	2.10	−150.01
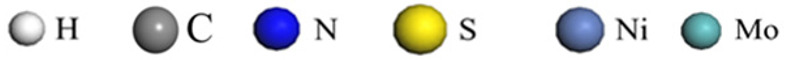

**Table 17 ijms-24-03044-t017:** Calculation methods and parameters.

Item	Parameter		
Function	General Gradient Approximation Perdew–Burke–Ernzerhof Function (GGA-RPBE) [37,51,52]		
Basis set	Double numerical plus polarization basis (DNP) [41,53]		
Electron spin	Open shell/unrestricted		
Symmetry	Asymmetry		
Self-consistent field density convergence (SCF)	2 × 10^−5^		
Thermal smearing	1 × 10^−3^ Hartree (Ha)		
Orbital cut-off	4.90 angstroms (Å)		
Core treatment	Effective core potentials (ECP)		
Dispersion correction	Grimme 06 [54]		
Exchange-correlation-dependent factor, s_6_	1.0 [55]		
Damping coefficient, d	20.0 [56]		
Grimme 6.0 Atomic dispersion [57]	Element	Interaction distance, R^0^	Dispersion coefficient C_6_
	H	1.001	1.451
	C	1.452	18.134
	N	1.397	12.748
	S	1.683	57.729
	Ni	1.562	111.943
	Mo	1.639	255.686

## Data Availability

The authors confirm that the data supporting the findings of this study are available within the article and its Appendix A.

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
