# Peer review of "Substituent Effects of the Nitrogen Heterocycle on Indole and Quinoline HDN Performance: A Combination of Experiments and Theoretical Study"

_ijms, 2023, doi:10.3390/ijms24033044_

Round 1

Reviewer 1 Report

In this valuable investigation by Cao, Yuan and co-workers the authors report the substituent effects of nitrogen heterocycle on indole and quinoline HDN performance. I am not an expert I this field, but to me, the research is designed appropriately, the methods adequately described, and the results clearly presented. I thus recommend acceptation of the article after the following modifications have been performed.

Editing of the English language and style is required.

Abstract: Whenever………active sites. Please check front size.

Each abbreviation must be defined as it appears in the text (CUS, LHSV, MHI, etc.).

Table 1 (second row). Formula and structural must be bolded. See also Table 2, 7th row; table 3, and 4, 6th row; table 12, 8th row. Table 14 and 15 are also misleading.

Homogenize size and template of tables (size of molecules and DFT drawings, text bolded or not bolded etc.).

Table 5: Post-hydrogenolysis.

Homogenize KJ.mol‒1 or KJ/mol

L89, 222, 255, 273, 290, 315, 329 must be bolded.

Elsewhere in the manuscript single bonds must be inserted via the insert button command and be the same size as for the ‒SH bond at L256.

Experimental section, L307. 2.0 of what acetic acid.

Reference section: references are not properly templated.

Author Response

Review #1

Question 1:Editing of the English language and style is required.

Answer 1:The English was polished by the English Editing of MDPI followed your proposal.

Question 2:Abstract: Whenever………active sites. Please check front size.

Answer 2:The front size of the abstract is unified by size 10.

Question 3:Each abbreviation must be defined as it appears in the text (CUS, LHSV, MHI, etc.).

Answer 3:The abbreviations have been defined and the emendations have been marked in the manuscript.

Question 4:Table 1 (second row). Formula and structural must be bolded. See also Table 2, 7th row; table 3, and 4, 6th row; table 12, 8th row. Table 14 and 15 are also misleading. Homogenize size and template of tables (size of molecules and DFT drawings, text bolded or not bolded etc.).

Answer 4:The formula of the tables have been unified. The size of the DFT drawings and the text has also been adjusted.

Question 5: Table 5, Post-hydrogenolysis.

Answer 5: We have corrected the spell.

Question 6: Homogenize KJ.mol‒1 or KJ/mol

Answer 6: The unit of energy has been unified into kJ·mol-1.

Question 7: L89, 222, 255, 273, 290, 315, 329 must be bolded.

Answer 7: The subtitles in the manuscript have been bolded.

Question 8: Elsewhere in the manuscript single bonds must be inserted via the insert button command and be the same size as for the ‒SH bond at L256.

Answer 8: The single bonds in the manuscript have been re-inserted.

Question 9: Experimental section, L307. 2.0 of what acetic acid.

Answer 9: The unit of acetic acid has been added.

Question 10: some references are not properly templated.

Answer 10: the format of the references have been adjusted.

Please see the attachment for the revised manuscript.

Reviewer 2 Report

The manuscript describes experimental and theoretical studies on catalytic hydrogenation of indole, quinoline and their methylated derivatives on sulfurized Ni-Mo-catalyst on aliminium oxide carrier. Based on analysis of hydrogenated products several reaction pathways, including heterocyclic ring depletion, which proceed on catalytic surface, were proposed. The effects of methyl groups on catalytic hydrogenation of indole and quinoline derivatives were investigated as well. The obtained experimental results are adequately described by computer models. The present paper gives clear mechanistic considerations for catalytic hydrogenation of indole and quinoline heterocycles. This underlines the significance of the work for catalysis and synthetic chemistry and makes it fit for publication in International Journal of Molecular Sciences (IJMS).

I should pay attention to the main drawbacks in the paper which reduce its quality:

1. The theoretical data and calculation are provided for Ni-Mo catalyst containing sulfur atoms. Despite sulfurization of catalyst is also described in Experimental Part, there is no mention of it in Results and Discussion (see Tables 2, 10, 12, 14). This must be indicated anyway to show that experimental conditions are in accordance with theoretical explanation.

2. It is not clear how the structure of hydrogenated compounds was elucidated. If the authors determined the formation of products by gas-liquid chromatography and MS-spectroscopy, the corresponding data for chromatograms and molecular ions must be given in Experimental Section and also provided by Supplementary Materials.

3. Some abbreviations are not diclosed in the text, such as CUS, LHSV.

4. Experimental data and Modeling  are written in Present Simple. On the contrary, Calculations are written in Past Simple. It is recommended to unify the grammatic style with preference to commonly used Past Simple.

Some additional mistakes:

5. Line 231: "converts to sp2 hybridization", correct to sp3.

6. Line 250: 3-M-QHQ, correct to 3-M-DHQ

7. Line 356 Anther, correct to  Another

8. Line 357 via should be marked as italic.

9. Table 8: In O-vinyl aniline and O-propenyl aniline, the "O"-index should be small or fully disclosed as "ortho".

10. Table 8: The structures of O-vinyl aniline and O-propenyl aniline must contain NH2 instead NH.

11. The 2D- and 3D-pictures in Table 8 are big enough to locate in the table cells.

Author Response

Review #2

Question 1: The theoretical data and calculation are provided for Ni-Mo catalyst containing sulfur atoms. Despite sulfurization of catalyst is also described in Experimental Part, there is no mention of it in Results and Discussion (see Tables 2, 10, 12, 14). This must be indicated anyway to show that experimental conditions are in accordance with theoretical explanation.

Answer 1: In the experimental parts, the sulfurized Ni-Mo-S/γ-Al2O3 has been emphasized several times to avoid Ambiguities.

Question 2: It is not clear how the structure of hydrogenated compounds was elucidated. If the authors determined the formation of products by gas-liquid chromatography and MS-spectroscopy, the corresponding data for chromatograms and molecular ions must be given in Experimental Section and also provided by Supplementary Materials.

Answer 2: The identification and quantification of HDN products has been reinforced in section 3. The chromatographic conditions and the peak positions for each products has been listed in the supplementary materials (at the end of the revised manuscript). The chromatograms for indole and Quinoline HDN products has also been provided in the supplementary materials.

Question 3: Some abbreviations are not diclosed in the text, such as CUS, LHSV.

Answer 3: The abbreviations have been defined and the emendations have been marked in the manuscript.

Question 4: Experimental data and Modeling are written in Present Simple. On the contrary, Calculations are written in Past Simple. It is recommended to unify the grammatic style with preference to commonly used Past Simple.

Answer 4: The English grammatic style were revised and polished by the English Editing of MDPI followed your proposal.

Question 5:

Some additional mistakes:

Line 231: "converts to sp2 hybridization", correct to sp3.

Line 250: 3-M-QHQ, correct to 3-M-DHQ.

Line 356 Anther, correct to  Another

Line 357 via should be marked as italic.

Table 8: In O-vinyl aniline and O-propenyl aniline, the "O"-index should be small or fully disclosed as "ortho".

Table 8: The structures of O-vinyl aniline and O-propenyl aniline must contain NH2 instead NH.

The 2D- and 3D-pictures in Table 8 are big enough to locate in the table cells.

Answer 5: These defects have been revised as your suggestion. 

Please see the attachment for the revised manuscript.

Round 2

Reviewer 2 Report

The authors corrected general drawbacks in the manuscript. It can be now published in IJMS.

I suggest the authors to provide all the experimental details in Supplemenatry Part when preparing future manuscripts.